# Nearly Half of Patients with Anaplastic Thyroid Cancer May Be Amenable to Immunotherapy

**DOI:** 10.3390/biomedicines12061304

**Published:** 2024-06-12

**Authors:** Beverley Chern, Diluka Pinto, Jeffrey Hy Lum, Rajeev Parameswaran

**Affiliations:** 1Division of Endocrine Surgery, National University Hospital Health System, Lower Kent Ridge Road, Singapore 119074, Singapore; bervychern95@gmail.com (B.C.); dilukpinto@yahoo.com (D.P.); 2Division of Surgery, Faculty of Medicine, University of Kelaniya, Thalagolla Road, Colombo P.O. Box 6, Sri Lanka; 3Department of Pathology, National University Hospital, Singapore 119074, Singapore; jeffrey_hy_lum@nuhs.edu.sg; 4Department of Surgery, Yong Loo Lin School of Medicine, Level 8, IE Kent Ridge Road, Singapore 119228, Singapore; 5NUS Centre for Cancer Research, Yong Loo Lin School of Medicine, National University of Singapore, Singapore 119077, Singapore

**Keywords:** anaplastic, poorly differentiated, thyroid, cancer, PD-L1, immunotherapy

## Abstract

Importance: Poorly differentiated cancer (PDC) and anaplastic thyroid cancer (ATC) have an aggressive course of disease with limited treatment options. The expression of programmed cell death ligand-1 (PD-L1) has been used to determine the responses of many cancers to immunotherapy. The aim of the study was to investigate the expression of PD-L1 in a cohort of patients with PDC and ATC to assess their suitability for immunotherapy. Data, settings, and participants: This study is a retrospective cohort review of patients treated for PDC and ATC treated at a tertiary referral institution during the period 2000–2020. PD-L1 22C3 pharmDx qualitative immunohistochemistry was performed on formalin-fixed, paraffin-embedded (FFPE) specimens of tumours to detect the presence of the PD-L1 protein. Main outcome measures: The percentage of tumours that were positive for PD-L1 immunohistochemistry and the PD-L1 protein expression as measured by using the Tumour Proportion Score (TPS). Secondary outcomes studied were the associations between demographic, clinicopathological, treatment and disease outcomes and PD-L1 expression. Results: Nineteen patients (12F:7M) with a mean age of 65.4 (±14.3 SD) years were diagnosed with PDC in 4 (21%) and fifteen were diagnosed with ATC (79%) during the study period. Fifteen (79%) patients underwent some form of surgery, with R0 resection achieved in only three of the fifteen (20%) patients. Overall, PD-L1 expression was seen in seven of the fifteen (47%) of the patients with ATC, with no positivity seen in the patients with PDC. PD-L1 expression had no impact on treatment modality and positive expression was not significantly associated with stage of disease, metastasis, or survival. Conclusion: Nearly half of patients with ATC express PD-L1 and may be amenable to immunotherapy with pembrolizumab.

## 1. Introduction

Thyroid cancers are the most common endocrine malignancies, and their incidence is rising globally. The most common cancers are differentiate thyroid cancers, namely papillary and follicular thyroid cancer, which have a slow rate of growth and a low potential for invasion and distant metastasis [1,2]. The outcomes of these cancers are excellent with conventional treatments. In contrast, poorly differentiated (PDC) and anaplastic thyroid cancers (ATC) grow rapidly and are fatal [3,4,5]. The incidence of these cancers is about 1–2% of all thyroid malignancies and most patients present with advanced disease or metastatic disease at presentation. Most patients with ATC rarely survive beyond 12 months even with the best treatment. The dismal prognosis necessitates the need to develop novel therapies for patients with PDC and ATC [2].

Recent advances in our understanding of the molecular pathogenesis of ATC have shown that a few genes play a role in the development and progression of the disease, mediated via a few immune-related genes, such as PPARGC1A [6], CD86, CD274, HAVCR2, and TDO2 [7]. A recent development in the treatment of cancer is immunotherapy using cell checkpoint inhibitors [8]. These drugs target the immune pathways which increases the body’s ability to recognize and destroy tumour cells. One such mediator is a cell surface glycoprotein called programmed death receptor 1 (PD-1) protein which is seen on some lymphocytes such as T cells, natural killer cells (NKs), dendritic cells (DCs) and some cancer cells [9].

Programmed death 1 (PD-1), also called B7 homolog1 (B7-H1) or cluster of differentiation CD274, is a transmembrane protein that plays an important role in mediating immunosuppression [10]. PD-1 that is located on the T-cells acts via the ligands PD-L1 (B7-H1) and PD-L2 (B7-DC), which are expressed either by tumour or stromal cells, or both [11,12]. When PD-L1 binds to T-lymphocytes, it inhibits the migration and proliferation of T cells, suppresses the release of cytotoxic mediators such as interleukins and inhibits tumour cell killing [13,14]. In many human cancer cells PD-L1 is upregulated to inhibit the anti-tumour T-cell responses, and this provides the rationale for immunotherapy using monoclonal antibodies targeting this pathway [11,14,15,16,17].

Anti-PD-L1 immunotherapy with pembrolizumab binds to the PD-1 receptor, thereby preventing interactions with PD-L1 and PD-L2 and augmenting the T-cell mediated immune response. The frequency of PD-L1 expression in thyroid cancers ranges between 23 to 88%. PD-L1 expression has been shown to correlate with a higher risk of recurrence and shortened disease-free survival in thyroid cancer patients [18,19]. The accurate detection of PD-L1-positive tumours in aggressive thyroid cancers can be important for identifying patients who may potentially benefit from anti-PD-L1 therapy. A recent study showed that pembrolizumab boosted the abscopal effects of radiotherapy by causing regression and tumour burden in anaplastic thyroid cancer [20]. The aim of the current study was to evaluate the expression of PD-L1 in ATC and PDC which may be useful for assessing if pembrolizumab may be effective in these aggressive cancers.

## 2. Materials and Methods

This is a retrospective study of patients diagnosed with PDC and ATC between the years 2000 and 2019 at a tertiary referral hospital. Clinicopathological data collected included age, gender, diagnosis, histopathological details (size, grade, invasion, subtype), imaging and fine needle biopsy results and details of surgery—tracheostomy, core biopsy, lobectomy, total or subtotal thyroidectomy with or without neck dissection, adjuvant therapies such as external beam radiotherapy (EBRT), chemotherapy or palliative therapies. Follow up and survival data were collected from the case notes if available. Overall survival (OS) was defined as the time to death from initial diagnosis. All samples used in the study were de-identified. The study was approved by the institutional review board (IRB 2020/00524).

### 2.1. IHC

For immunohistochemistry, representative sections of the tumour were prepared from paraffin embedded blocks and representative samples were identified by two experienced endocrine pathologists. Then, 4 µm thick sections were prepared and stained with haematoxylin and eosin. Additional sections were cut and immunostained with anti–PD-L1 22C3 mouse monoclonal primary antibody and negative controls (normal thyroid tissue from within the same tumour sample) using the Dako Auto-stainer Link 48 platform (Agilent Technologies, Singapore) as described in the PD-L1 IHC 22C3 pharmDx [21].

It has been shown that the Dako PD-L1 IHC 22C3 assay is a sensitive, specific, precise and robust assay, which has high value clinical utility, though it is intended for the detection of PD-L1 protein in formalin-fixed, paraffin-embedded (FFPE) samples of non-small cell lung cancer (NSCLC) tissue [21]. However, there have been a few reports published in the literature of using PD-L1 immunohistochemistry to guide targeted therapies in ATC treatment [22,23,24].

The expression of PD-L1 detected by the antibodies was interpreted blindly by a pathologist. PD-L1 protein expression was measured by using Tumour Proportion Score (TPS), which is the percentage of viable tumour cells showing partial or complete membrane staining at any intensity. A dichotomous scoring system was used: a. percentage of tumour cells showing PD-L1 positive staining. A score of 5% or more was categorized as ‘PD-L1-positive’ and a score of less than 5% as ‘PD-L1-negative’ as shown in other studies [13,14,16]; b. intensity of staining positivity was defined as complete circumferential or partial cell membrane staining of viable tumour cells with 1+ to 3+ intensity. Tumour-associated immune cells and cytoplasmic staining were excluded from PD-L1 scoring.

### 2.2. Statistical Analysis

The statistical analysis of the data was performed using SPSS 25.0 (IBM Corporation, New York, NY, USA) and the associations of PD-L1 staining with the clinicopathological variables were compared using rates and proportions with Chi-square (X^2^ test), whereas continuous data were evaluated using the *t*-test. To assess the impact of PD-L1 with survival, Kaplan–Meier survival curves were generated and compared using the log-rank test. Multivariate analysis was performed using Cox regression analysis to calculate predictors affecting OS with calculation of hazard ratio. Overall, a *p*-value of less than 0.05 was considered significant.

## 3. Results

A total of thirty-nine patients were diagnosed with the condition over the time-period, but twenty patients were excluded from the study, as shown in Figure 1. The mean age of the nineteen patients (12F:7M) in the cohort was 65.4 (±14.3) years, and PDC in four (21%) were diagnosed with PDC and fifteen with ATC (79%) during the study period. The mean size of the tumours was 4.8 (±1.84; range 1.5 to 9.3) cm. Of the nineteen patients, fifteen (79%) underwent surgery (twelve with total thyroidectomy and three with a hemithyroidectomy), and four (21%) underwent a core biopsy for diagnosis. The demographic and clinicopathological details are shown in Table 1.

Seventeen (89%) of the patients presented with a rapidly enlarging mass that was associated with pressure symptoms (dysphagia in ten, dysphonia in four and dyspnoea/orthopnoea in one patient), and two patients presented with metastasis (one with a spinal lesion and the other with an abdominal wall mass). Four patients had a previous history of papillary thyroid cancer for which they had undergone a total thyroidectomy, followed by radioiodine ablation. None of the patients had a family history of thyroid cancer or history of childhood radiation exposure. The thyroid function tests were normal in all except two of the patients who had a suppressed TSH following a thyroidectomy for PTC. All patients underwent imaging with an ultrasound scan and a computerized tomography (CT) scan; two patients also underwent magnetic resonance imaging (MRI) and two also underwent positron emission tomography (FDG-PET) preoperatively.

In the fifteen patients who underwent surgery, an R0 resection was achieved only in three patients, an R1 resection in nine and an R2 resection in three patients. All fifteen patients also had some form of lymph nodal surgery, with six undergoing a central neck dissection and the other five undergoing both central and lateral neck dissection. Recurrences following surgery were seen in eleven of the fifteen (73%) patients, of whom seven (64%) had recurrences in the thyroid bed and four (36%) in the nodal basins after a mean follow up of 24 months (range of 2 to 138 months). Surgical intervention for the recurrences was only possible in three of the eleven (27%), with two patients undergoing a total laryngectomy and one of these patients also underwent selective neck dissection.

Palliative treatments were offered to patients who were not suitable candidates for surgery, both at index presentation and following inoperable recurrences. Nine of the nineteen (47%) patients received-high dose I-131radioiodine ablation (the tumours were positive for Tg expression), three of the nineteen (16%) received palliative external beam radiotherapy (EBRT) and one patient received multimodal chemotherapy (5%). Acute airway obstruction was seen in six of the nineteen patients (32%), with five being treated with a palliative tracheostomy and one patient with tracheal stenting. In two patients with ATC, a percutaneous endoscopic gastrostomy (PEG) was placed.

The pathology showed ATC in fifteen of the nineteen patients (79%) and PDC in four (21%) patients. In the four patients with PDC, aggressive variants included the insular variant in two, and the trabecular and sclerosing variant in one patient. PD-L1 expression was seen in seven of the fifteen (47%) patients with ATC and none of the PDC patients (Figure 2). The TPS of the positive tumours is show in Table 2 and the score ranged between 60 and 90% in the anaplastic thyroid cancer patients. PD-L1 expression did not correlate with any of the clinicopathological parameters evaluated, including those of treatment modality, stage of disease and metastasis (Table 3). The expression of PD-L1 did not correlate with poorer survival, as shown by the survival curve (log rank test 0.128) (Figure 3). The cohort’s mortality was 84%, with all fifteen patients with ATC and one patient with PDC, succumbing to progressive disease after a mean follow up period of 24 months (range of 2 to 138 months).

The PD-L1 staining percentage reported in the literature for ATC and PDC ranges between 13 and 100%, as shown in Table 4.

## 4. Discussion

The main findings of this study are as follows: about half of anaplastic thyroid cancers express PD-L1, making them amenable to anti-PD-L1 therapy with pembrolizumab; PD-L1 expression was not a predictor of poor prognosis or survival in this cohort; PD-L1 expression is absent in poorly differentiated cancers; and finally, anaplastic thyroid cancer continues to have dismal survival rates, with no curative therapy available to date. The prognosis of patients with PDC and ATC is generally poor due to the aggressive biology of disease where they present with locally invasive disease, early metastasis and rapid progression of the disease [5,34,35]. The absence of any effective therapy and an improved molecular profiling of these cancers have led researchers to look for alternative therapies, especially immunotherapy [31,32].

Anaplastic thyroid cancer may arise de novo or from differentiated thyroid cancer. In this cohort, anaplastic thyroid cancer developed in 20% of the patients with history of differentiated thyroid cancer. The molecular features of ATC frequently show mutations of TP53, TERT, RAS, BRAF and aberrations in the WNT, RAS/Raf/MEK/ERK and PI3 K/AKT/mTOR signalling pathways [36,37,38,39]. Given the high mutational burden (TMB), tumour-associated macrophages (TAMs) and higher PD-L1 expression, these tumours may, therefore, be amenable to immunotherapy with anti-PD-L1 inhibitors [40,41]. A few studies have reported PD-L1 expression in various thyroid cancers, including papillary, follicular and anaplastic thyroid cancer [26,27,28,33,42,43,44,45]. In our cohort, 47 percent of the ATC tumours expressed PD-L1, and the expression did not show any impact on any of the parameters studied unlike other published series [27].

Anti-PD-L1 immunotherapy with pembrolizumab, which is an immune checkpoint inhibitor, targets the PD-1 on immune cells. Studies have shown that an increased expression of PD-L1 or high TMB levels may show a beneficial response to therapy with pembrolizumab in non-thyroid cancers [46,47]. In a phase I study in patients with advanced thyroid cancer who did not respond to standard therapies, pembrolizumab improved both clinical responses and survival [48]. Treatment with pembrolizumab in combination with tyrosine kinase inhibitors in patients with progressive disease was shown to be as effective as salvage therapy in a series of 12 ATC patients [33]. In the study by Iyer et al. a clinical benefit was seen in 75% of patients whereby the overall survival increased by 6.9 months and responses to anti-PD1 immunotherapy were seen irrespective of the PD-L1 expression on the tumour. However, in another study of 16 ATC patients, an increased PD-L1 expression of more than 33% was associated with shorter survival [29].

In this cohort, PD-L1 expression was lacking in 12 tumours, and this poses the question as to whether there would be any role for antiPD-L1 therapy despite the lack of expression. Patients with a poor response to anti-PD-L1 therapy show absent PD-L1 expression, with a lack of T-cell infiltrates. In the absence of T-cells, the tumour cells may not express PD-L1 and therefore supports the argument that anti-PD-L1 therapy may be not effective in these patients. In one study, despite the high PD-L1 and TMB, the survival impact was quite low [49]. The hypothesis of there being a lack of response was attributed to rapid rate of growth of the tumours and the time needed for the drug to elicit a response [32]. Conversely, in ATC with low TMB and high microsatellite instability (MSI), a partial response to therapy with pembrolizumab was shown [50].

Immunotherapy may be offered as a single agent therapy or in combination with the multikinase inhibitor lenvatinib [51]. Only a handful of studies have been reported using immunotherapy with pembrolizumab, livonumab, spartalizumab and Ipilimumab as the sole agents and the results from the studies were disappointing [41,49,52]. Pembrolizumab may be a treatment option for a subset of patients despite the low response rates, especially as a neoadjuvant therapy in combination with chemotherapy and selective BRAF and MEK inhibitor agents such as dabrafenib/trametinib to enable surgical resection [53], or in very locally advanced disease involving the aerodigestive system [54]. In the cohort reported on here, 32% of the patients with ATC presented with aerodigestive system involvement and a total laryngectomy was performed in two patients. It is possible that neoadjuvant treatment with pembrolizumab may have ameliorated some of the symptoms and less radical surgery could have been performed followed by chemoradiotherapy in these patients.

Most studies that have reported on the use of immunotherapy in ATC as combination therapy with lenvatinib [31,32,33,55]. There are four studies with a combined total of 45 patients which have shown an overall response rate ranging between 52 and 66%, a mean overall survival between 6.3 and 18.5 months and a one-year survival rate of a maximum of 50%. The response rate to combination therapy (lenvatinib with pembrolizumab) shows that is has a much better efficacy than monotherapy with pembrolizumab. The combination therapy may be used in pre-treated patients [32,33] or as a first-line therapy [31]. Evidence showing whether therapy lenvatinib with pembrolizumab should be initiated before or after other systemic therapies is lacking.

In this study, 63 percent of the tumours were PD-L1-negative. One of the dilemmas is that the true negative rates of PD-L1 negativity may not be the true negative but rather a false negative that may result from using an archival sample where the samples may have some degradation of PD-L1 preventing us from making a true assessment. It is also possible that the tumour samples may not have an invasive front where the T-cells may have induced PD-L1 expression or the tumours may have been very heterogeneous, with some areas showing expression and others possibly not [56]. It is, therefore, important that PD-L1 expression be studied on biopsy samples collected recently or with an adequate amount of tissue to make a reasonable analysis. It is not uncommon to find areas of necrosis in ATC and PDC, and when these areas are sampled, there may not be any representative tumour tissue to evaluate PD-L1 expression. Studies in some human cancers have shown that tumour necrosis correlates with PD-L1 expression and indicate an aggressive phenotype with a poor prognosis [57,58], possibly due to the upregulation of hypoxia-inducible factor-1α 67 and tumour necrosis-α [59,60]. However, in this cohort, there was no correlation between tumour necrosis and PD-L1 expression.

There are a few limitations to the study. The number of patients in this cohort was quite small, but given the rarity of these tumours, especially those in which surgical intervention can be undertaken, must be take into consideration. The presented results have biases due to the retrospective nature of the study. The mutational burden of the tumours was not studied to study their correlation with PD-L1 expression in this cohort, which is a major limitation.

In conclusion, based on the immunohistochemical profile, PD-L1 therapy may be feasible in patients with ATC. However, this requires evaluation in clinical trials to assess the response to anti-PD-L1 therapy in aggressive thyroid cancers such as ATC.

## Figures and Tables

**Figure 1 biomedicines-12-01304-f001:**
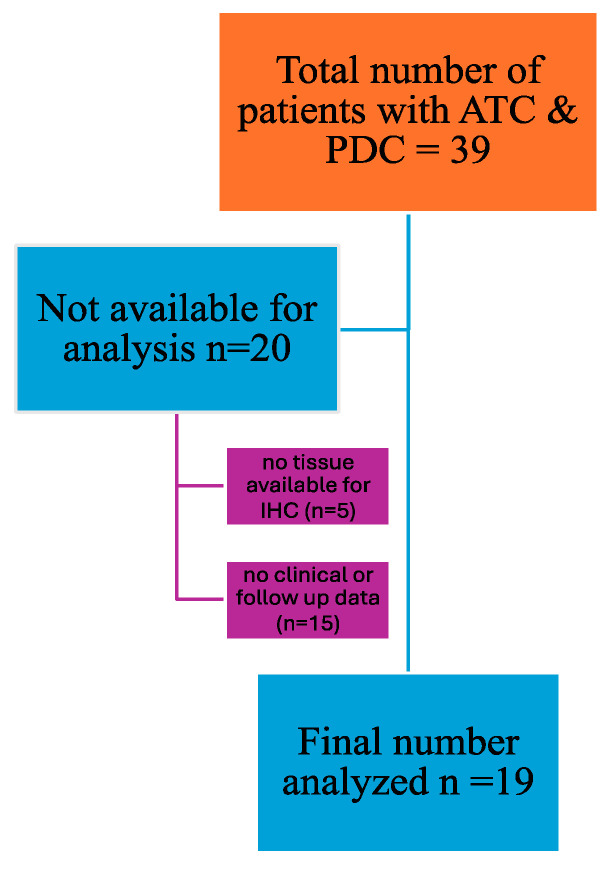
Flow chart of patients included in the study for PD-L1 immunohistochemistry.

**Figure 2 biomedicines-12-01304-f002:**
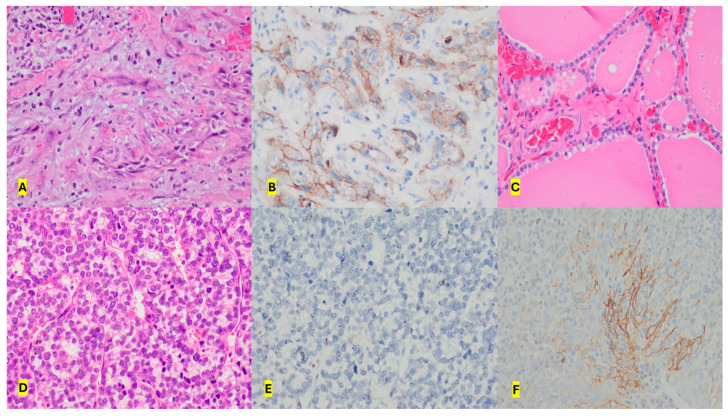
(**A**) Case of anaplastic thyroid cancer on H&E staining. (**B**) Positive PD-L1 staining of the same case. (**C**) Negative control of normal thyroid sample from the same tumour sample. (**D**) Case of poorly differentiated thyroid cancer on H&E staining. (**E**) Negative PD-L1 staining of the poorly differentiated cancer. (**F**) Positive control of PD-L1 staining in tonsillar tissue sample. All samples are shown with a 40× magnification (high power).

**Figure 3 biomedicines-12-01304-f003:**
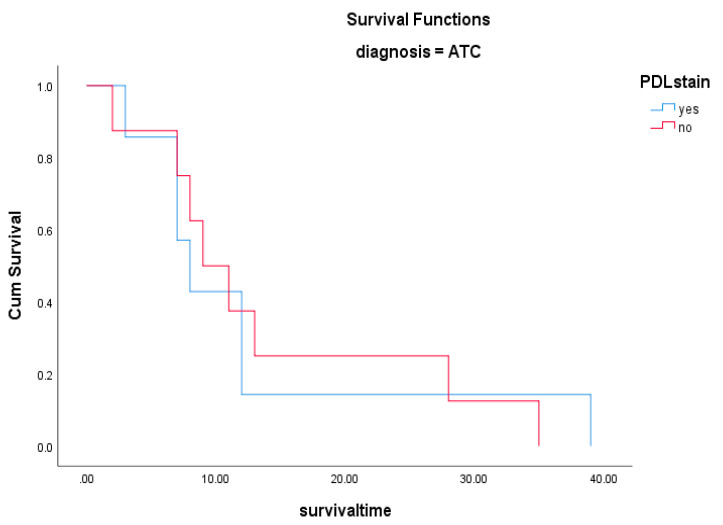
Kaplan–Meier survival curve showing no impact of PD-L1 expression on survival in ATC patients (*p* = 0.684).

**Table 1 biomedicines-12-01304-t001:** Demographic and clinicopathological details of the patients in the study.

Parameter	Number
Age (mean/range)	65.4 (41–86)
Gender (F:M)	12:7
Tumour type	
ATC	15 (79%)
PDC	4 (21%)
Surgery	
Biopsy only	4 (21%)
Hemithyroidectomy	3 (16%)
Total thyroidectomy	12 (53%)
Pathological features (in resected specimen)	
Lymphovascular invasion	13 (68%)
Capsular invasion	13 (68%)
Soft tissue invasion	12 (63%)
Gross invasion	12 (63%)
Resection margins (in resected specimen)	
R0	3 (20%)
R1	9 (60%)
R2	3 (20%)
Distant metastasis	11 (58%)
Cohort mortality	17 (89%)

**Table 2 biomedicines-12-01304-t002:** The TPS score of the 19 patients who underwent PD-L1 staining.

Patient ID	Diagnosis	TPS	Intensity of Staining
1	ATC	0	none
2	ATC	80%	+++
3	ATC	0	none
4	ATC	0	none
5	ATC	0	none
6	ATC	60%	++
7	ATC	70%	++
8	PDC	0	none
9	ATC	0	none
10	ATC	90%	+++
11	ATC	90%	+++
12	PDC	0	none
13	ATC	80%	+++
14	ATC	0	none
15	ATC	90%	+++
16	PDC	0	none
17	ATC	0	none
18	PDC	0	none
19	ATC	0	none

Abbreviations: ATC—anaplastic thyroid cancer; PDC—poorly differentiated cancer. TPS—tumour proportion score. ++/+++—intensity of staining.

**Table 3 biomedicines-12-01304-t003:** Correlation of PD-L1 staining with the clinicopathological parameters.

Parameter (N = 19)		Positive PD-L1 Staining	*p*-Value *
Gender	Male (n = 7)	2 (28%)	0.568
	Female (n = 12)	12 (42%)	
Diagnosis	ATC (n = 15)	7 (37%)	0.086
	PDC (n = 4)	0 (0%)	
Lymph node status	Positive (n = 10)	3(30%)	0.656
	Negative (n = 9)	4 (44%)	
Lymphovascular invasion	Positive (n = 10)	3 (30%)	0.68
	Negative (n = 5)	1 (20%)	
Capsular invasion	Positive (n = 10)	4 (40%)	0.099
	Negative (n = 5)	0 (0%)	
Extrathyroidal extension	Positive (n = 10)	3 (30%)	0.68
	Negative (n = 5)	1 (20%)	
Gross extension	Positive (n = 10)	3 (30%)	0.68
	Negative (n = 5)	1 (20%)	
Mortality	Yes (n = 16)	7 (44%)	0.149
	No (n = 3)	0 (0%)	

* Pearson Chi-square’s test *p* < 0.05; PD-L1—programmed death-ligand-1.

**Table 4 biomedicines-12-01304-t004:** Clinical studies of PD-L1 expression in ATC/PDC published in the literature.

Study	Number of Patients with ATC/PDC	PD-L1 Positivity N (%)	Type of Antibody Used for IHC
Wu et al., 2015 [25]	13	3 (23)	Mouse MAb
Bastman et al., 2016 [26]	10	7 (70)	Rabbit MAb
Ahn et al. 2017 [27]	15	2 (13)	Rabbit MAb
Zwaenepoel et al., 2017 [28]	49	21 (43)	Rabbit MAb
Chintakutlawar et al., 2017 [29]	16	13 (81)	Rabbit MAb
Rosenbaum et al., 2018 [30]	28	7 (25)	Rabbit MAb
Soll et al., 2024 [31]	5	5 (100)	Rabbit MAb
Dierks et al., 2021 [32]	8	7 (83)	Rabbit MAb
Iyer et al., 2018 [33]	12	10 (83)	Rabbit MAb
Total	156	65 (45)	

Abbreviations: ATC—anaplastic thyroid cancer; PDC—poorly differentiated cancer; IHC—immunohistochemistry.

## Data Availability

Data will not be shared publicly but will be available at reasonable request from the senior author. The data are not publicly available due to institutional policies.

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
