# Peer review of "Nearly Half of Patients with Anaplastic Thyroid Cancer May Be Amenable to Immunotherapy"

_biomedicines, 2024, doi:10.3390/biomedicines12061304_

Round 1

Reviewer 1 Report (Previous Reviewer 1)

Comments and Suggestions for Authors

My previous comments have been adequately addressed. The paper has reached a high enough priority to be acceptable for publication

Author Response

Thank you

Reviewer 2 Report (Previous Reviewer 2)

Comments and Suggestions for Authors

The manuscript titled “Nearly half of patients with anaplastic thyroid cancer may be amenable to immunotherapy”, by Chern et al., evaluates the immunoexpression of PD-L1 in a series of PDC and ATC cases to assess potential patients’ selection for immunotherapy. As long as the total number of cases in the study is small, these thyroid carcinomas are rare, so any study on ATC and PDC is worth attention.

But, while the main scope of the manuscript is well described in the introduction section and discussed in the discussion section, most of the results section is about the clinicopathological data of the patients, treatment procedures, their complications, and their outcomes. And there is no experimental data about treatment with pembrolizumab. Furthermore, in lines 276-278, it is written: “In this study, clinical benefit was seen in 75% of patients whereby the overall survival increased by 6.9 months and responses to anti-PD1 immunotherapy was seen irrespective of the PD-L1 expression on the tumor.” I could not find any data in the results section of the presented manuscript that could support such conclusions.

Therefore, there is a huge discrepancy between the presented results and the rest of the paper, and the conclusions are not experimentally supported. It is mandatory to focus the results section on PD-L1 expression and its associations with clinocopathological data of the patients, the disease's outcome, and complications, and to paraphrase the title and the conclusions. Immunotherapy with pembrolizumab should be excluded from the conclusion, as you did not perform experiments that support such findings. You may only suggest the potential application of your results. Besides this major flow, here are some minors:

1.        Lines 45, 47, 48, 51, 52 – references needed.

2.        Line 64-65 – “The function of PD-1 is mainly in the periphery where it interacts with T-cells…..”. PD-1 does not interact with T cells; PD-1 is on T- cells. PD-L1 is on the tumor cell, and then PD-1 binds PD-L1. Please, correct.

3.        Line 98 - What was the negative control? Please, add.

4.        Line 109-113 – Why was the dichotomous scoring system described? As I could notice, only a percentage of positive cells was applied.

5.        Lines 203-205 – Associations of PD-L1 immunoexpression with tested clinicopathological data should be presented in a separate table.

6.        Figure 2 presents PD-L1 IHC staining in ATC and control benign tissue. The diagnosis of control is missing (is it follicular thyroid adenoma?) and it is necessary to add the number of samples stained and the exact diagnosis in the material and methods section. Optionally, you may present PD-L1 IHC staining of PDC here (instead of a benign thyroid sample).

7.        According to your results, PDC does not express PD-L1. Therefore, it is meaningless to present it in the Kaplan-Meier survival curve. Therefore, I suggest you exclude PDC cases from this analysis and then repeat the testing only on ATC cases. The statistics may be changed.

8.        Please, after the abbreviation introduction, always use the abbreviation afterwards in the text (ATC) and keep consistency with abbreviations throughout the manuscript (PD-L1 or PDL-1).

9.        Line 267 – “The PD-L1 staining percentage reported in the literature for ATC and PDC ranges between 1 to 64% as shown in Table 4.” But, in Table 4, the range of PD-L1 positivity is 13–100%. Please explain or correct. In addition, the phrase “as shown in Table” should not be placed in the discussion section.

10.   Table 4 - An explanation of abbreviations should be placed under the table (No, N, ATC, PDC).

Comments on the Quality of English Language

minor English editing is needed

Author Response

The manuscript titled “Nearly half of patients with anaplastic thyroid cancer may be amenable to immunotherapy”, by Chern et al., evaluates the immunoexpression of PD-L1 in a series of PDC and ATC cases to assess potential patients’ selection for immunotherapy. As long as the total number of cases in the study is small, these thyroid carcinomas are rare, so any study on ATC and PDC is worth attention.

But, while the main scope of the manuscript is well described in the introduction section and discussed in the discussion section, most of the results section is about the clinicopathological data of the patients, treatment procedures, their complications, and their outcomes. And there is no experimental data about treatment with pembrolizumab. Furthermore, in lines 276-278, it is written: “In this study, clinical benefit was seen in 75% of patients whereby the overall survival increased by 6.9 months and responses to anti-PD1 immunotherapy was seen irrespective of the PD-L1 expression on the tumor.” I could not find any data in the results section of the presented manuscript that could support such conclusions.

We thank the reviewer for the observation. The statement in this study, was actually rereferring to the study by Iyer et al and we have amended the statement to make it clear.

Therefore, there is a huge discrepancy between the presented results and the rest of the paper, and the conclusions are not experimentally supported. It is mandatory to focus the results section on PD-L1 expression and its associations with clinocopathological data of the patients, the disease's outcome, and complications, and to paraphrase the title and the conclusions. Immunotherapy with pembrolizumab should be excluded from the conclusion, as you did not perform experiments that support such findings. You may only suggest the potential application of your results. Besides this major flow, here are some minors:

  1. Lines 45, 47, 48, 51, 52 – references needed.

References added

  1. Line 64-65 – “The function of PD-1 is mainly in the periphery where it interacts with T-cells…..”. PD-1 does not interact with T cells; PD-1 is on T- cells. PD-L1 is on the tumor cell, and then PD-1 binds PD-L1. Please, correct.

Thank the reviewer for the observation. Have amended the statement.

  1. Line 98 - What was the negative control? Please, add.

Have added (benign thyroid tissue within the same tumour samples)

  1. Line 109-113 – Why was the dichotomous scoring system described? As I could notice, only a percentage of positive cells was applied.

Both the percentage of cells and staining intensity was measured. Have added. The staining intensity to the table as well to reflect the dichotomous system.

  1. Lines 203-205 – Associations of PD-L1 immunoexpression with tested clinicopathological data should be presented in a separate table.

Have added the necessary table.

  1. Figure 2 presents PD-L1 IHC staining in ATC and control benign tissue. The diagnosis of control is missing (is it follicular thyroid adenoma?) and it is necessary to add the number of samples stained and the exact diagnosis in the material and methods section. Optionally, you may present PD-L1 IHC staining of PDC here (instead of a benign thyroid sample).

Thank the reviewer for the valuable comment. The controls used is benign thyroid tissue within the same tumour samples for the negative controls and tonsillar tissue as positive controls.

  1. According to your results, PDC does not express PD-L1. Therefore, it is meaningless to present it in the Kaplan-Meier survival curve. Therefore, I suggest you exclude PDC cases from this analysis and then repeat the testing only on ATC cases. The statistics may be changed.

Have removed PDC from the Kaplan Meier survival curve and statistics corrected accordingly as well.

  1. Please, after the abbreviation introduction, always use the abbreviation afterwards in the text (ATC) and keep consistency with abbreviations throughout the manuscript (PD-L1 or PDL-1).

Have made the necessary changes in the manuscript.

  1. Line 267 – “The PD-L1 staining percentage reported in the literature for ATC and PDC ranges between 1 to 64% as shown in Table 4.” But, in Table 4, the range of PD-L1 positivity is 13–100%. Please explain or correct. In addition, the phrase “as shown in Table” should not be placed in the discussion section.

Have made the necessary corrections.

  1. Table 4 - An explanation of abbreviations should be placed under the table (No, N, ATC, PDC).

Have made the necessary corrections.

Round 2

Reviewer 2 Report (Previous Reviewer 2)

Comments and Suggestions for Authors

Dear authors,

The revised version of the manuscript titled “Nearly half of patients with anaplastic thyroid cancer may be amenable to immunotherapy”, by Chern et al., has been sufficiently improved, and with a few minor corrections, I recommend it for publication:

  1. Abstract, line 27: The explanation of the “IHC” acronym should be added.
  2. Abstract, line 29: There are two “the”. Please delete one.
  3. On Page 4, lines 183–184, it is written: ”Complications that were seen in patients following thyroid surgery is shown in Table 2”. But Table 2 presents the TPS score of the 19 patients who underwent PD-L1 staining. As I could have noticed, the complications are presented in the body text of the manuscript, not in the table. Please correct accordingly.

Author Response

Dear authors,

The revised version of the manuscript titled “Nearly half of patients with anaplastic thyroid cancer may be amenable to immunotherapy”, by Chern et al., has been sufficiently improved, and with a few minor corrections, I recommend it for publication:

  1. Abstract, line 27: The explanation of the “IHC” acronym should be added.

Amended

  1. Abstract, line 29: There are two “the”. Please delete one.

Amended

  1. On Page 4, lines 183–184, it is written: ”Complications that were seen in patients following thyroid surgery is shown in Table 2”. But Table 2 presents the TPS score of the 19 patients who underwent PD-L1 staining. As I could have noticed, the complications are presented in the body text of the manuscript, not in the table. Please correct accordingly.

That statement was not needed and hence removed.

We would like to thank the reviewers for their valuable insight and comments which made the article worth publishing.

Regards

Rajeev Parameswaran on behalf of all the authors

This manuscript is a resubmission of an earlier submission. The following is a list of the peer review reports and author responses from that submission.

Round 1

Reviewer 1 Report

Comments and Suggestions for Authors

This study investigates the expression of programmed cell death ligand-1 (PD-L1) in poorly differentiated cancer (PDC) and anaplastic thyroid cancer (ATC), both of which have limited treatment options and aggressive disease courses. PD-L1 expression was observed in 47% of ATC cases, while PDC patients showed no positivity. The study suggests that nearly half of ATC patients may benefit from immunotherapy with pembrolizumab, highlighting the potential relevance of PD-L1 expression as a marker for treatment suitability in this context.

Major comments:

- There are few recent data concerning the potential role of few immune-related genes in the initiation and progression of ATC. A short description should be reported in the Introduction. Please see and cite: PMID: 32720497, PMID: 36530988

- It is widely acknowledged that radioiodine plays no significant role in the treatment of ATC. I find it quite surprising that, in your series, 9 out of 19 (47%) patients received high-dose radioiodine. You have only 4 patients with PDC. This means that at least 5 patients with ATC have been treated with 131-I. Could you provide an explanation for this? Additionally, I would like to confirm the accuracy of the histological diagnosis of ATC in these cases.

- The authors should better underline the main novelty of the present manuscript compared to previous studies.

- All figures and their corresponding legends appear to be missing.

Minor comments:

- The spelled-out form of each acronym should be indicated on first use.

- Please, check the use of acronyms. An abbreviation should be used only if the term appears at least 4-5 times in the main text. If the term is used only few times it should not be abbreviated.

- Although the manuscript is generally well presented, there are few typing errors.

- As a supplementary file, the authors downloaded the “PD-L1 IHC 22C3 pharmDx Interpretation Manual, by Agilent Dako.” I question the appropriateness of this file based on the instructions for authors.

Reviewer 2 Report

Comments and Suggestions for Authors

Dear authors,

The manuscript “Nearly half of patients with anaplastic thyroid cancer may be amenable to immunotherapy”, biomedicines-2849924, describes and discusses the IHC staining of PD-L1 in ATC patients in order to use it as an excluding criterion for immunotherapy with pembrolizumab. As long as ATC is a rare type of thyroid carcinoma, the sample size of this study is too small to make any general conclusion. Furthermore, the study presented in this way has many flaws, of which a lot are major. Therefore, unfortunately, I cannot recommend it for publication.

Major:

1.    The study's conclusion needs to be revised. It could not be stated from the supplied results. There is no testing group for pembrolizumab, no in vitro or in vivo investigations, and no clinical trials were conducted. I suggest you rewrite it, describe the interesting noticing, represent it via its potential usefulness, and discuss it in the discussion section.

2.      English editing is needed, because the manuscript has a lot of confusing or grammatically incorrect sentences. Please read the whole manuscript carefully and correct it.

3.      The assay PD-L1 IHC 22C3 pharmDx, used in your study, is a qualitative immunohistochemical assay using Monoclonal Mouse Anti-PD-L1, Clone 22C3 intended for the detection of PD-L1 protein in formalin-fixed, paraffin-embedded (FFPE) samples of non-small cell lung cancer (NSCLC) tissue. This should all be noted in the Material and Method section of the manuscript. Furthermore, how did you validate your results as you were testing the expression of PD-L1 in thyroid tissue samples?

4.      The results of the statistical analysis are missing.

5.      I could not find any figures in the manuscript.

6.      Lines 139-141: “The cohort mortality was 89%, 139 with all 15 patients with ATC and 2 patients with PDC, succumbing to progressive disease 140 after a mean follow up period of 24 months (range 2 to 138 months).” The mortality should be presented in dependence on PD-L1 expression via Kaplan-Meier survival curves.

Minor:

1.      The abstract should be structured (prepared) according to the journal’s guidelines for the authors (Introduction, Material and Metod, Results, Conclusion).

2.      The number of cases included in the study should be clearly presented in the Material and Methods section of the abstract.

3.      Lines 36-40: The references are missing. Each sentence presented in lines (36-40) should be referenced.

4.      On the other hand, the presented reference list is too long, so it should be shortened.

5.      The references in the text should be cited in brackets.

6.      Line 51-53: The subcellular location, role, and function of PD-L1 are not explained well in the introduction section of the manuscript. I suggest you shift (and adapt) some parts from the discussion section into the introduction part (e.g., lines 163-170). Additionally, the basic mechanism of immunotherapy (with pembrolizumab) should be explained in the introduction.

7.      Lines 59-60: The references 11 and 12 are studies on papillary thyroid carcinoma (PTC) patients, not thyroid cancer patients in general. Please correct it and describe some results for ATC cases here.

8.      Line 103 and line 104: please correct the symbol ([email protected], [email protected])

9.      Line 101–135: Most of the text presented in the results section is not the experimental results of this study. It’s clinical data that is not correlated to PD-L1 expression. Therefore, it should be shifted into the Material and Methods section. In general, the results of the study are poorly presented. As I could notice, only the text written in 5 lines (from line 136 to line 141) is describing your experimental data.

10.   Each abbreviation in each table should be described under the table.

11.   Please use the same abbreviation throughout the manuscript (PD-L1 or PDL-1).

12.   Line 238 – Table 2 presents only complications during the surgery. It does not show the correlation of complications with PD-L1 expression. Please correct. Or, this correlation may be interesting, so you may correlate complications with PD-L1 expression and present it in Table 2.

Comments on the Quality of English Language

English quality is poor. The manuscript has a lot of confusing or grammatically incorrect sentences.

Reviewer 3 Report

Comments and Suggestions for Authors

The authors present their work on PD-L1 expression in a series of PDC and ATC. As themselves accepted by the authors, the number of cases is small. The results section provides information on things that are not even part of the objectives, e.g., complications of surgery. Table 3 needs to be presented better. For e.g., in the second column, what is implied by 'yes' and 'no'? Do they mean that they were PD-L1-positive and negative, respectively. In third column, does the term 'nil' encompass all scores that are less than 5% or refers to TPS zero?

References are incomplete. Some of the publications missed include 30762153, 32364844, 32611231, 34093774, 34475850, 35574031, 37006068, 32936917.

Most part of the discussion section is irrelevant to the objectives of the work.

For the results to be meaningful, it is essential to document correlation of PD-L1 status with response to immunotherapy. She has not been performed. Also molecular profile and TMB influence PD-L1 expression. These have not been assessed.

Comments on the Quality of English Language

There are grammatical errors that need to be catered too.

Reviewer 4 Report

Comments and Suggestions for Authors

            COMMENTS  

The manuscript titled “Nearly Half of Patients with Anaplastic Thyroid Cancer May Be Amenable to Immunotherapy” of Beverley Chern et al., reports an investigation on the role of programmed cell death ligand-1 (PD-L1) in immunotherapy of patients suffering from poorly differentiated cancer (PDC) and anaplastic thyroid cancer (ATC).

Two were the aims of this investigation. Firstly, to evaluate the immunoexpression of PD-L1 in a cohort of PDC and ATC patients to assess their suitability for immunotherapy. Secondly, to check if PD-L1 expressions were associated with demographic, clinicopathological, treatment and disease outcomes of these patients.

To perform this retrospective study, immunohistochemistry methods were employed to detect PD-L1 expressions on formalin-fixed, paraffin-embedded specimens of thyroid cancerous lesions. By using Tumor Proportion Score, the percentage of immunoreactive cancers and cellular PD-L1 immunostaining were reported. To assess the impact of PD-L1 with survival, statistical tests were performed.

The results of this study showed that PD-L1 immunoexpressions were in 7 of 15 (47%) ATC and in none of PDC (100%) patients. Further, PD-L1 expression did not correlate with any of the parameters evaluated including that of treatment modality, stage of disease, metastasis, and survival.

In conclusion, PD-L1 immunoexpressions may be useful to select ATC patients for therapy with pembrolizumab.

Abstract:

Abstract section isn’t adequately describing this study in the section of results.

Minor:

Lines 15-16: this sentence should be revised.

Lines 19-23: these sentences should be revised.

Lines 26-27: this sentence should be revised.

Lines 27-28: this sentence should be omitted.

Introduction:     

This section is adequately describing only the premises of this study.

Major:

The aim of this study is not expressed in this section.

Minor:

The number of references should be placed in brackets.

Materials and Methods:           

This section provides sufficient information.

However, isn’t information about the immunohistochemistry procedures. Further, there aren’t enough information about the number of patients and their clinical-pathological features. There I’d suggest authors to move the lines 101-126 from “Results” to this section.

Minor:

Lines 102-104: this sentence should be revised.

Results:

This section provides detailed information. However,

Minor:

See comments in “Introduction” section.

Discussion:

The comments of discussion are appropriate for this investigation.

Conclusions:      

The conclusions are appropriate.

Tables: give a helpful visual representation of study.

Major

Figures: there aren’t figures to give a valid visual representation of study.

Bibliography/References:

References are adequate.

Decision:

This study may be accepted for publication after major revisions.

Comments on the Quality of English Language

Minor editing of English language required